# Contrastive Adapters for Foundation Model Group Robustness

**Michael Zhang and Christopher Ré**
Department of Computer Science
Stanford University
{mzhang, chrismre}@cs.stanford.edu

## Abstract

While large pretrained foundation models (FMs) have shown remarkable zero-shot classification robustness to dataset-level distribution shifts, their robustness to sub-population or group shifts is relatively underexplored. We study this problem, and find that foundation models such as CLIP may not be robust to various group shifts. Across 9 robustness benchmarks, zero-shot classification with their embeddings results in gaps of up to 80.7 percentage points (pp) between average and worst-group accuracy. Unfortunately, existing methods to improve robustness require retraining, which can be prohibitively expensive on large foundation models. We also find that efficient ways to improve model inference (*e.g.*, via adapters, lightweight networks that transform FM embeddings) do not consistently improve and can sometimes *hurt* group robustness compared to zero-shot. We therefore develop an adapter training strategy to effectively and efficiently improve FM group robustness. Our motivating observation is that while poor robustness results from groups in the same class being embedded far apart in the foundation model "embedding space," standard adapter training may not actually bring these points closer together. We thus propose contrastive adapting, which contrastively trains adapters to bring sample embeddings close to both their ground-truth class embeddings *and* same-class *sample* embeddings. Across the 9 robustness benchmarks, contrastive adapting consistently improves group robustness, raising worst-group accuracy by 8.5 to 56.0 pp over zero-shot. Our approach is also efficient, doing so without any FM finetuning and only a fixed set of FM embeddings. On popular benchmarks such as Waterbirds and CelebA, this leads to worst-group accuracy comparable to state-of-the-art methods, while only training $\leq 1\%$ of the model parameters.

## 1 Introduction

Foundation models (FMs)—large pretrained models trained on massive datasets—offer an exciting new paradigm for deep learning. Recent works have shown that without any finetuning, foundation models can generalize well to various datasets [11, 36, 59, 69] and exhibit impressive robustness to certain distribution shifts [42, 76]. Under this zero-shot paradigm, practitioners can avoid training task-specific models, and instead use FM embeddings for efficient and effective inference.

However, an underexplored question is how robust this zero-shot inference is to "group shifts," distribution shifts between subpopulations or meaningful groups in data. Prior works have established that *group robustness—i.e.* performing well on all groups—is a fundamental and real-world challenge for modern deep learning [5, 12, 40, 51, 55, 66, 71]. Yet most prior foundation model evaluations focus on overall or average performance [42, 59, 76]; few works consider FM accuracy across groups.

In this work, we thus study foundation model group robustness. We motivate this problem by first showing that foundation models can have poor zero-shot group robustness. Evaluating 11 foundation

36th Conference on Neural Information Processing Systems (NeurIPS 2022).

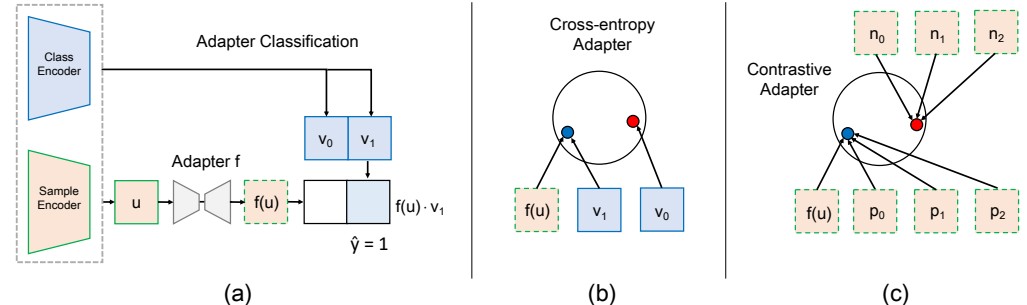

Figure 1: (a) Adapter classification with FM embeddings. Adapters learn transformations to align sample embeddings to ground-truth class embeddings. (b) Cross-entropy loss encourages alignment between class embeddings [22]. (c) Contrastive adapting adds alignment between sample embeddings.

models across 9 robustness benchmarks, we find they achieve up to an 80.7 percentage point (pp) gap between average and worst group accuracy, only classifying 6.0% of worst-group samples correctly.

We therefore aim to improve FM group robustness. This poses several challenges and open questions. First, while improving group robustness in machine learning is well-studied, existing robustness methods require retraining one (and often more than one) entire models [1, 16, 39, 47, 51, 65, 71, 72, 79]. This can be prohibitively expensive for foundation models due to their size and scale, raising the question of whether we can make these models more robust without any retraining or finetuning. Second, for zero-shot classification, many practitioners may also only access foundation model outputs or embeddings (*e.g.*, via APIs[1]). To improve robustness, ideal solutions should only require pretrained FM embeddings. However, these same embeddings lead to poor zero-shot robustness, raising the question of if they even encode the information needed to classify all groups correctly.

Motivated by these challenges and questions, we study effective *and* efficient solutions for better FM group robustness. As a baseline, we first find that while efficient methods to improve FM inference— such as training linear probes [42, 59] and adapters [22, 33] on top of FM embeddings—can improve group robustness over zero-shot (reducing the gap by up to 50.2 pp on representative benchmarks), they fail to do so consistently, and can *hurt* robustness. They reduce worst-group accuracy by up to 37.9 pp, and increase the accuracy gap by up to 74.9 pp. To reason about this inconsistency, we note that poor zero-shot robustness results when FMs embed same-class samples in different groups "far apart" in embedding space. While adapter training achieves higher robustness than linear probing, we find settings where it still fails to close this distance, *e.g.*, if training data is group-imbalanced.

To then handle these scenarios and consistently improve group robustness over zero-shot, we propose *contrastive adapting*, a simple adapter training method that places greater emphasis on bringing these initially "far apart" points together. For each task, we first use foundation models to compute embeddings for each training sample and class. We then train adapters—small bottleneck MLPs—on these embeddings. Like prior work [22], these adapters take sample embeddings as inputs, and output transformed embeddings with greater cosine similarity to their ground-truth class embeddings. However, the key difference is that contrastive adapting also applies a supervised contrastive loss over other *sample* embeddings. Specifically, we provide a way to "pull together" far apart sample embeddings in the same class, and "push apart" nearby sample embeddings in different classes.

In our experiments, we validate that contrastive adapting effectively and efficiently improves FM group robustness. First, across all 9 robustness benchmarks, we find contrastive adapting consistently improves worst-group accuracy over zero-shot (by 8.5 to 56.0 pp), using no training group labels and only training MLPs with 0.1% to 0.3% of the original FM parameters. Then, on a representative set of benchmarks with various group shifts and training data group sizes, we find contrastive adapting can substantially outperform prior adapter training strategies, and outperforms other approaches that only use fixed FM embeddings (achieving up to 12.4 pp higher worst-group accuracy than the next best method on average). Finally, beyond just improving FM robustness, we find contrastive adapting also achieves effective and efficient group robust classification in general. We achieve near state-of-the-art (SoTA) or SoTA worst-group accuracy on popular robustness benchmarks with only 1.0% of the trainable parameters (*e.g.*, improving 0.2 pp over the prior SoTA [52] on CelebA [48]).

---

[1]https://beta.openai.com/docs/introduction., https://studio.ai21.com/docs/, https://docs.cohere.ai/

In summary, we find that while FM zero-shot classification may not be group-robust, we can significantly improve robustness without any finetuning. This suggests the information to classify groups is frequently in their pretrained embeddings; we may just need proper methods to extract it.

## 2   Related Work

Our work builds on (i) methods to improve group robustness, and (ii) methods to improve foundation model inference without accessing or finetuning their original weights. We briefly describe these works here, and include an expanded discussion in Appendix D.

**Improving group robustness.** Many works aim to improve group robustness. If training group labels are known, prior methods often balance group sizes during training, via sample balancing [17, 28, 34, 39], importance weighting [13, 68], or robust optimization [2, 65]. We do not assume training group labels. With these assumptions, a common approach first trains a model with empirical risk minimization (ERM), before using this model's predictions to infer groups. Methods then train a second robust model with sample balancing [47, 51] or robust optimization [16, 52, 71] using inferred group labels, or representation learning to learn similar representations for groups in the same class [79]. While effective at improving group robustness, these solutions require training one (and often more than one) models. This can make applying them to foundation models prohibitively expensive.

**Improving foundation model inference efficiently.** Other prior works improve foundation model downstream performance, without having to finetune or update original model weights. *Prompt tuning* optimizes the inputs of a FM while keeping the original model weights frozen. Optimizing either text [43, 45, 83, 84] or image [3, 77] inputs can improve a frozen foundation model's downstream task accuracy. However, doing so can require multiple passes through the foundation model, which may become expensive in certain situations (*e.g.*, interacting with the model via a commercial API). Another paradigm adds small trainable parameters to the original model, either within its layers or on top of its embeddings. These include linear probes (linear classifiers) [59] and adapters (small bottleneck MLPs) [33, 57, 58, 60]. Recently, Kumar et al. [42], Wortsman et al. [76] propose methods with linear probes to improve robustness after finetuning to out-of-distribution (OOD) shifts [30, 32, 62, 74]. Gao et al. [22] train adapters on pretrained embeddings to improve average downstream accuracy. We focus on *group shifts* within a dataset. We also show standard adapter training can hurt group robustness, and propose alternatives to consistently improve group robustness.

## 3   Problem

In Section 3.1, we first describe the group robustness problem setting. In Section 3.2, we illustrate this problem with foundation models. We show that zero-shot classification with foundation models, and existing baseline approaches to improve downstream inference, can result in poor group robustness.

### 3.1   Preliminaries: group robustness and task setup

We emphasize robustness to distribution shifts between groups in this work. For setup, we follow prior works [40, 47, 65, 71] that alternatively describe the phenomenon as *hidden stratification* [71] or *subpopulation shift* [40]. For some task, we have $N$ samples $\{(x_i, y_i, g_i)\}_{i=1}^N$, with sample features or inputs $x_i \in \mathcal{X}$, class labels $y_i \in \mathcal{Y}$, and group labels $g_i \in \mathcal{G}$. Let $C = |\mathcal{Y}|$ be the number of classes. We use $g_i$ to indicate the group that each sample belongs in, but do not observe group labels during training. Distribution shifts may occur between samples in different groups but the same class.

Every sample $(x_i, y_i, g_i)$ is drawn from some unknown joint distribution $P$. Let $P_g$ be the specific distribution conditioned on $g$ for any $g \in \mathcal{G}$. For classification loss $\ell : \mathcal{Y} \times \mathcal{Y} \mapsto \mathbb{R}$ and classifier $f_\theta : \mathcal{X} \mapsto \mathcal{Y}$, we want $f_\theta$ to be accurate, *i.e.* achieving low average error:

$$\mathcal{L}_{\text{avg}}(f_\theta) := \mathbb{E}_{(x,y,g) \sim P}[\ell(f_\theta(x), y)] \tag{1}$$

and *group robust*, *i.e.*, achieving a small gap between its average error and its worst-group error:

$$\mathcal{L}_{\text{wg}}(f_\theta) := \max_{g \in \mathcal{G}} \mathbb{E}_{(x,y,g) \sim P_g}[\ell(f_\theta(x), y)] \tag{2}$$

Different from domain generalization or OOD evaluation settings [31, 32, 44, 64, 82], we observe each data group in training, validation, and test splits. However, standard training via empirical risk minimization (ERM) can still lead to poor test set group robustness because training groups may be imbalanced [65, 71, 79]. Here, foundation models are *not trained* on the training data, but we show that zero-shot classification with foundation models can still result in poor group robustness.

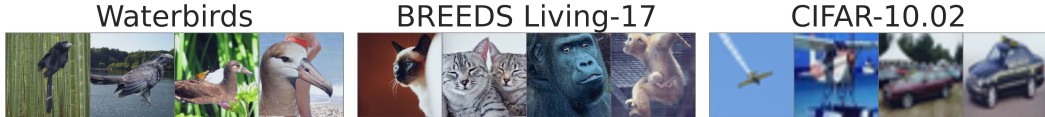

Figure 2: Samples of different group shifts for robust evaluation (2 classes, 2 groups per class shown).

### 3.2 Empirical findings of poor foundation model group robustness

To motivate the rest of this work, we now demonstrate the group robustness problem with foundation models. We first describe different natural group shifts for evaluation. We next detail primary baseline approaches. We finally summarize our findings after evaluating these baselines on 11 popular foundation models across 9 standard group robustness benchmarks used in prior work [40, 49, 61, 65, 67]. We present four representative scenarios based on training data assumptions and group robustness outcome. Critically, we find that zero-shot classification with foundation models may result in poor group robustness. We also find that baseline methods to improve downstream transfer do not consistently improve group robustness, and can make group robustness worse.

**Dataset group shifts.** We benchmark methods on the following sources of group shift (Figure 2):

- **Spurious confounders.** We evaluate across groups which may or may not carry spurious confounders—input features predictive for some, but not all groups in a class. For example, in Waterbirds [65, 75], a water background is a confounder for the `waterbirds` class.

- **Subclass variance.** We evaluate across groups which are different fine-grained subclasses. For example, in BREEDS Living-17 [67], the `ape` class includes images of gibbons and gorillas.

- **Data source variance.** We evaluate across groups which are the same class but sourced from different datasets. For example, we set up the CIFAR-10.02 dataset by combining CIFAR-10 [41] and CIFAR-10.2 [49]. The `airplanes` class contains samples from both datasets.

**Baseline methods.** To evaluate foundation model group robustness, we consider the following baseline methods. Following prior work [20, 36, 46, 50, 59], for all approaches we first compute $N$ sample embeddings and $C$ class embeddings using a foundation model. With foundation model embedding dimension $D$, let $u_n \in \mathbb{R}^D$ be a sample embedding and $c_n \in \mathbb{R}^D$ be a class embedding.

- **Zero-shot classification** [59]: We classify each sample via the nearest class embedding to its sample embedding $u_n$. Specifically, we compute the class-wise logits for each sample $x_n$ as

$$f_\theta(x_n; \tau) = \hat{W}^\top \hat{u}_n / \tau \tag{3}$$

where $\hat{u}_n = u_n / \|u_n\|$ is the ($\ell$-2) normalized sample embedding of $x_n$, $\hat{W} \in \mathbb{R}^{D \times C}$ is a matrix whose columns are the normalized class embeddings $\{\hat{v}_c\}_{c=1}^C$, and $\tau$ is a temperature parameter. The highest class logit corresponds to the nearest neighbor and largest dot product. As standard, for class embeddings we convert each class name to a natural language prompt, *e.g.*, "photo of a `[class name]`", and feed the tokenized prompt to a foundation model's text encoder. As in prior work [59], we engineer class prompts by trying several templates. We defer details to Appendix A.2, such as optimal templates (Table 11) and a list of all templates tried (Table 20).

- **Linear Probe** [59, 76]: We train a linear classifier on top of training data sample embeddings. Specifically, with classifier $f_\theta(u) = W^\top u$, we update the weights $W \in \mathbb{R}^{D \times C}$ with a cross-entropy loss applied over training data sample embeddings $\{u_n\}_{n=1}^N$ and labels $\{y_n\}_{n=1}^N$.

- **Adapter** [22, 60]: We train a single 2-layer bottleneck multilayer perception (MLP) to output transformed sample embeddings, which we use instead of the original sample embeddings to classify with in the zero-shot procedure above. Specifically, with adapter hidden-layer dimension $H$, ReLU activation function $\sigma$, and adapter weights $\phi = [W_1, W_2]$—where $W_1 \in \mathbb{R}^{D \times H}$ is a linear down-projection and $W_2 \in \mathbb{R}^{H \times D}$ a linear up-projection—we compute "adapted" embeddings

$$f_\phi(u) = W_2^\top \sigma \left( W_1^\top u \right) \tag{4}$$

We classify samples with the zero-shot class matrix $\hat{W}$, temperature $\tau$, and normalized adapted embeddings $\hat{f}_\phi(u) = f_\theta(u) / \|f_\phi(u)\|$. The final outputs are given by $f_\theta(u; \hat{W}, \tau) = \hat{W}^\top \hat{f}_\phi(u) / \tau$. Like with linear probes, we update $\phi$ with a cross-entropy loss using training data labels $\{y_n\}_{n=1}^N$ and a softmax over the dot product-computed logits as class-wise probabilities.

For evaluation, we train both linear probes and adapters with standard empirical risk minimization (ERM), which aims to minimize the empirical risk: $\hat{\mathcal{L}}(f_\theta) = \frac{1}{N} \sum_{n=1}^N \ell(f_\theta(u_n), y_n)$.

Table 1: Baseline worst-group (WG) and average (Avg) accuracies with zero-shot classification, linear probes, and adapters. Best metric **in bold**. While training linear probes and adapters can improve group robustness (reducing the worst-group versus average accuracy gap by 57.4 pp on BREEDS Living-17), it can also result in poorer robustness (in red), increasing the gap by 74.9 pp on CelebA.

| Method | Waterbirds | | | CelebA | | | BREEDS Living-17 | | | CIFAR-10.02 | | |
|---|---|---|---|---|---|---|---|---|---|---|---|---|
| Accuracy (%) | WG | Avg | Gap | WG | Avg | Gap | WG | Avg | Gap | WG | Avg | Gap |
| Zero-shot | 36.6 | 92.2 | 55.6 | **74.0** | 81.9 | **7.9** | 6.0 | 86.7 | 80.7 | 39.1 | 69.9 | 30.8 |
| Linear Probe | 7.9 | 93.5 | 85.6 | 11.9 | 94.7 | 82.8 | 53.3 | 90.8 | 37.5 | 51.3 | 77.7 | 26.4 |
| Adapter | **60.8** | 96.0 | **35.2** | 36.1 | 94.2 | 58.1 | **70.7** | 94.0 | 23.3 | **68.8** | 86.0 | **17.2** |

Table 2: Representative outcomes for improving group robustness.

| | | Class-wise Group Size | | | Improved Group Robustness? | |
|---|---|---|---|---|---|---|
| Example Dataset | Group Shift | Largest | Smallest | Balanced? | Linear Probe | Adapter |
| Waterbirds | Confounder | 1057 | 56 | ✗ | ✗ | ✓ |
| CelebA | Confounder | 22880 | 1387 | ✗ | ✗ | ✗ |
| BREEDS Living-17 | Subclass | 1076 | 1009 | ✓ | ✓ | ✓ |
| CIFAR-10.02 | Data source | 4039 | 431 | ✗ | ✓ | ✓ |

**Discussion and representative outcomes.** In Table 1, we report worst-group and average accuracies along with their corresponding gaps on four representative group robustness datasets, using zero-shot classification, linear probes, and adapters on CLIP ResNet-50 embeddings. We select datasets to report based on training data setup and group robustness outcome, where we find that (i) the relative group size ratios, (ii) the type of group shift, and (iii) the choice of adapter or linear probe influences group robustness improvements. We note descriptive characteristics and outcomes in Table 2, and summarize three main takeaways below. Appendix A contains results for all datasets and models.

1. **Foundation model zero-shot classification may not be group robust**: Across datasets, we find that zero-shot classification with CLIP ResNet-50 embeddings can achieve 7.9 to 80.7 pp gaps between worst-group and average accuracy. Worryingly, poor group robustness is accompanied by high *average* error (from 69.9% to 92.9%), the usual metric for evaluating zero-shot classification. This further supports the importance of improving group robustness.

2. **Efficient baselines do not consistently improve robustness**: We find that while previously proposed linear probes and adapters are efficient ways to improve accuracy on downstream tasks, these benefits do not consistently carry over to improving group robustness.

   - When training data is balanced, both linear probes and adapters can substantially improve group robustness and worst-group accuracy (reducing the robustness gap by 43.2 and 54.7 pp respectively on BREEDS Living-17). However, when minority groups are rare, in some instances, approaches can hurt group robustness. On CelebA, adapters and linear probes increase the gap by 50.2 and 74.9 pp, and reduce worst-group accuracy by 37.9 and 62.1 pp.

3. **We can improve group robustness with only foundation model embeddings**: Our positive results suggest that poor zero-shot classification may not be because sample embeddings lack the information required to classify groups correctly. Rather, we may just require the right training strategies to learn how to better classify by this information.

Altogether, takeaways 1 and 2 motivate the need for methods to effectively improve robustness in the foundation model setting. Takeaway 3 suggests we can make progress on this problem.

## 4 Method

Having established the group robustness problem in Section 3, we now propose a simple contrastive adapter training strategy to improve group robustness. In Section 4.1, we setup our approach by identifying possible sources of limitation with standard adapter training. In Section 4.2, we then use these insights to propose a simple yet effective approach that counteracts these limitations.

### 4.1 Understanding prior limitations via embedding metrics

To guide a first-step strategy for improving robustness, we first outline high-level reasoning for why zero-shot and ERM-trained adapters fail to classify groups correctly. Recall that a key property of group robust classification is that all sample embeddings belonging to the same class should embed closer to their ground-truth class embedding than any other class embedding. If zero-shot classification for a specific class is accurate on average but not group robust, then in the pretrained

foundation model embedding space there exists groups that embed "close" to their ground-truth class embedding, and groups in the same class that embed "far away" (measured via cosine similarity). One way to interpret standard adapter training with FM embeddings via ERM is that it aims to bring these initially far apart sample embeddings closer to their ground-truth class embedding. Restating the standard sample cross-entropy loss with adapters makes this clear as an InfoNCE loss [14, 56]:

$$\ell(f_\theta(u), y) = -\log \frac{\exp(\hat{f}_\theta(u)^\top \hat{v}/\tau)}{\sum_{c=1}^{C} \exp(\hat{f}_\theta(u)^\top \hat{v}_c/\tau)} \tag{5}$$

with sample embedding $u$ as an anchor, class embedding $v$ of ground-truth $y$ as a single positive, and the other $C-1$ class embeddings as negatives. Via ERM of the sample cross-entropy loss, adapters thus bring zero-shot-incorrect anchors closer to their class embedding positives (minimizing Eq. 5).

However, in Section 3 we found this loss works in some scenarios but not others. Intuitively, Eq. 5 can fail to bring samples closer to their correct class embedding (*e.g.*, on CelebA). To find additional ways to bring points together, we hypothesize that poor robustness also accompanies poor similarity between *sample embeddings* from different groups but the same class. We verify this in Figure 3 by empirically measuring the average pairwise cosine similarity and group alignment loss $\mathcal{L}_{\text{align}}$ [79]—which measures the pairwise Euclidean distance—between sample embeddings in the same class but different groups. We compare these metrics with embeddings computed with trained adapters and the initial foundation model embeddings, and find that higher worst-group accuracy corresponds to higher cosine similarity and lower alignment loss between groups in the same class.

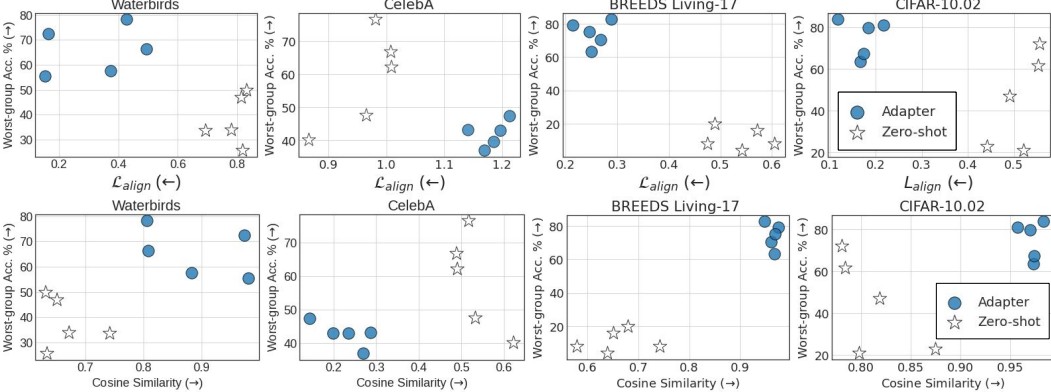

Figure 3: Across CLIP model architectures, cosine similarity and alignment loss between groups of the same class tracks worst-group error. Notably, training ERM adapters may fail to move these metrics in the desired direction, which corresponds with poorer robustness (*e.g.*, on CelebA).

## 4.2 Approach: Contrastive Adapting

To improve robustness, we therefore propose to more effectively bring far away samples together by introducing greater training signal via other *sample* embeddings. Instead of limiting ourselves to a single class embedding positive and a limited set of $C-1$ negatives, we expand our positives by including *sample* embeddings for points in the same class far away from the anchors among pretrained embeddings (*e.g.*, likely in different groups). We expand our negatives with sample embeddings from different classes. Following prior work [23, 79] that finds sampling *hard negatives* beneficial for robust contrastive learning, we also use the computed foundation model sample embeddings to sample negatives from points nearest to the anchors but in different classes. As the number of training data points $N$ is often much larger than the number of classes $C$, these choices are further supported by prior work suggesting more positives and negatives are beneficial for contrastive learning [38, 63]. In practice, contrastive adapting is simple to implement with three components:

- **Foundation model embedding and prediction**: We compute FM embeddings over labeled training data. To guide sampling, we collect zero-shot predictions over this data.
- **Contrastive sampling**: For each class, we identify an "anchor" sample embedding $u \in \mathcal{U}$ that zero-shot predicts incorrectly, and $P$ "positive" sample embeddings $\mathcal{P}(u)$ that zero-shot classifies correctly. We do this as a heuristic for finding samples "far apart" in the FM embedding space, so pushing them together improves robustness over zero-shot. We also identify $M$ hard "negative" sample embeddings $\mathcal{M}(u)$ by computing the nearest neighbors to the anchors in different classes, using cosine similarity between the sample embeddings.

- **Training objective**: We use a supervised contrastive loss [38] with the sample embeddings, *i.e.*

$$\ell_{\text{con}}^{\text{sup}}(f_\theta(u)) = \frac{-1}{P} \sum_{p \in \mathcal{P}(u)} \log \frac{\exp(\hat{f}_\theta(u)^\top \hat{f}_\theta(p)/\tau)}{\exp(\hat{f}_\theta(u)^\top \hat{f}_\theta(p)/\tau) + \sum_{m \in \mathcal{M}(u)} \exp(\hat{f}_\theta(u)^\top \hat{f}_\theta(m)/\tau)} \quad (6)$$

We also use a standard cross-entropy loss over minibatches of sample embeddings and their class embeddings. This aims to keep adapted embeddings close to their ground-truth class embeddings. To avoid undoing Eq. 6 and push "far away" points closer to their ground-truth class embeddings, we upsample the number of zero-shot incorrect samples to equal the number of zero-shot correct samples in each minibatch. We thus use contrastive supervision from class and sample embeddings.

**Robust generalization with adapters.** The contrastive loss in Equation 6 is also supported by recent results suggesting that minimizing the class-wise alignment loss $\mathcal{L}_{\text{align}}$ helps bound the worst-group versus average error gap for that class (*cf*. Thm 3.1, Zhang et al. [79]). The bound however scales with the Lipschitz constant of the neural network, and upper bounds for estimating this constant can grow with the size of the network [19, 73]. However, as our adapters are small 2-layer MLPs, estimates of this constant suggest we can obtain better generalization with fewer training samples [25, 37, 53, 78]. In Section 5.3, we later show this corresponds to better data efficiency.

## 5 Experiments

We now validate that contrastive adapting enables effective and efficient group robustness. First, in Section 5.1, we evaluate the effectiveness of contrastive adapting against efficient methods to improve FM inference. We study whether the approach consistently improves worst-group accuracy and group robustness over zero-shot classification, how contrastive adapting compares against other efficient methods that only require pretrained model embeddings, and whether contrastive adapting scales to a variety of pretrained model architectures. Next, in Section 5.2, we shed further light on contrastive adapting's performance by studying the importance of its individual components, ablating the contrastive objective and sampling strategy. Finally, in Section 5.3, we study the efficiency of contrastive adapting against effective group robustness approaches. We find that the prior robustness gains are not only relative to other efficient FM training methods; contrastive adapting also enables state-of-the-art robustness on popular benchmarks, but with greater parameter and data efficiency.

### 5.1 Robustness comparison for efficient foundation model methods

To first judge the effectiveness of contrastive adapting, we evaluate the method across the same set of initial robustness benchmarks and foundation model architectures discussed in Section 3. As in prior group robustness evaluation, we do not assume training groups labels, but do assume group labels in validation data for hyperparameter tuning and model selection [40]. We include experimental details for all models and hyperparameters in Appendix C.

As baselines, we compare against zero-shot classification [59], ERM linear probing [42, 59], and ERM adapter training [22]. We also compare against recent methods designed to improve downstream transfer in related settings, while similarly only requiring pretrained model embeddings:

- **Weight space ensembling (WiSE-FT)** [76], which first trains a linear classifier with standard ERM, and then ensembles the classifier outputs with the initial zero-shot predictions. While proposed for both training linear classifiers and finetuning the original weights of a foundation model, we focus on the linear classifier version for fair comparison in our setting.
- **Deep feature reweighting (DFR)** [39], which first trains a linear probe on embeddings computed from a pretrained model over group-balanced data. As we do not assume training group labels, we first infer groups using zero-shot classification with foundation model embeddings. As in prior work [47, 79], we treat the incorrect and correctly classified samples as proxies for different groups.

Finally, if we have validation group labels, we plausibly know what groups are in the test data. We thus also compare against **group-informed prompting** (Group Prompt ZS), which performs zero-shot classification using prompts with group information (*e.g.*, "a waterbird on a land background").

**Consistent robustness improvements over zero-shot.** In Figure 4 we report contrastive adapting's relative gains in worst-group accuracy over zero-shot classification on all 9 robustness benchmarks. Unlike prior adapter training approaches, contrastive adapting consistently improves group robustness over zero-shot classification, achieving 8.5 to 56.0 pp higher worst-group accuracy.

Table 3: Evaluation of methods for improving group robustness of CLIP models. Across representative benchmarks and CLIP models, contrastive adapters consistently improve worst-group accuracy over zero-shot classification (by 10.2 to 76.0 pp). **1st** / 2nd best worst-group (WG) and robustness gaps **bolded** / underlined.

| | Method / Acc. (%) | Waterbirds | | | CelebA | | | BREEDS Living-17 | | | CIFAR-10.02 | | |
|---|---|---|---|---|---|---|---|---|---|---|---|---|---|
| | | WG | Avg | Gap | WG | Avg | Gap | WG | Avg | Gap | WG | Avg | Gap |
| CLIP ResNet-50 | Zero-shot (ZS) | 36.6 | 92.2 | 55.6 | 74.0 | 81.9 | 7.9 | 6.0 | 86.7 | 80.7 | 39.1 | 69.9 | 30.8 |
| | Group Prompt ZS | 55.9 | 87.8 | 31.9 | 70.8 | 82.6 | 11.8 | 30.0 | 90.6 | 60.6 | N/A | N/A | N/A |
| | ERM Linear Probe | 7.9 | 93.5 | 85.6 | 11.9 | 94.7 | 82.8 | 53.3 | 90.8 | 37.5 | 51.3 | 77.7 | 26.4 |
| | ERM Adapter | 60.8 | 96.0 | 35.2 | 36.1 | 94.2 | 58.1 | **70.7** | 94.0 | **23.3** | **68.8** | 86.0 | **17.2** |
| | WiSE-FT | 49.8 | 91.0 | 41.2 | 85.6 | 88.6 | 3.0 | 53.3 | 90.8 | 37.5 | 58.2 | 79.1 | 20.9 |
| | DFR (Subsample) | 63.9 | 91.8 | 27.9 | 76.9 | 92.5 | 15.6 | 46.7 | 89.4 | 42.7 | 45.0 | 75.0 | 30.0 |
| | DFR (Upsample) | 51.3 | 92.4 | 41.1 | 89.6 | 91.8 | 2.2 | 44.0 | 86.4 | 42.4 | 38.5 | 77.9 | 39.4 |
| | **Contrastive Adapter** | **83.7** | 89.4 | **5.7** | **90.0** | 90.7 | **0.7** | 62.0 | 90.9 | 28.9 | 60.7 | 80.9 | 20.2 |
| CLIP ViT-L/14 | Zero-shot (ZS) | 25.7 | 87.3 | 61.6 | 62.1 | 71.9 | 9.8 | 4.0 | 86.6 | 82.6 | 72.0 | 93.2 | 21.2 |
| | Group Prompt ZS | 27.4 | 85.5 | 58.1 | 72.4 | 81.8 | 9.4 | 48.0 | 96.6 | 48.6 | N/A | N/A | N/A |
| | ERM Linear Probe | 65.9 | 97.6 | 31.7 | 28.3 | 94.7 | 66.4 | **84.0** | 98.6 | 14.6 | **87.5** | 96.1 | **8.6** |
| | ERM Adapter | 78.4 | 97.8 | 19.4 | 36.7 | 94.2 | 57.5 | 82.8 | 98.2 | 15.5 | 87.0 | 96.9 | 9.9 |
| | WiSE-FT | 65.9 | 97.6 | 31.7 | 80.0 | 87.4 | 7.4 | **84.0** | 98.6 | 14.6 | **87.5** | 97.0 | 9.5 |
| | DFR (Subsample) | 51.9 | 95.7 | 43.8 | 76.3 | 92.1 | 15.8 | **84.0** | 98.5 | **14.5** | 85.5 | 96.6 | 11.1 |
| | DFR (Upsample) | 65.9 | 96.1 | 30.2 | 83.7 | 91.2 | 7.5 | 78.7 | 97.3 | 18.6 | 72.5 | 93.8 | 21.3 |
| | **Contrastive Adapter** | **86.9** | 96.2 | **9.3** | **84.6** | 90.4 | **5.8** | 80.0 | 97.5 | 17.5 | 82.2 | 96.1 | 13.9 |

Table 4: On the Waterbirds dataset, contrastive adapters consistently improve group robustness across various vision-language large pretrained models (CLIP [59], CLOOB [20]) and backbones (ResNets and ViTs).

| Accuracy (%) | CLOOB RN-50 | | | CLOOB RN-50x4 | | | CLIP RN-101 | | | CLIP ViT-B/32 | | | CLIP ViT-B/16 | | |
|---|---|---|---|---|---|---|---|---|---|---|---|---|---|---|---|
| | WG | Avg | Gap | WG | Avg | Gap | WG | Avg | Gap | WG | Avg | Gap | WG | Avg | Gap |
| Zero-shot | 41.6 | 60.4 | 18.8 | 24.1 | 51.1 | 27 | 33.6 | **90.0** | 56.4 | 47.0 | **88.8** | 41.8 | 34.0 | 88.1 | 54.1 |
| Contrastive Adapter | **83.0** | **86.8** | **3.8** | **85.8** | **88.5** | **2.7** | **82.0** | 86.0 | **4.0** | **80.7** | 84.2 | **3.5** | **83.1** | **90.9** | **7.8** |

**Representative dataset evaluation.** In Table 3 we compare contrastive adapting to other lightweight methods for improving robustness. We evaluate with group-imbalanced and balanced training data across spurious confounder, subclass, and data source group shifts, using CLIP ResNet-50 (RN-50) and CLIP ViT-L/14 models. On average, contrastive adapters raise worst-group accuracy by 12.4 and 4.1 pp over the next best methods on CLIP RN-50 and ViT-L/14 models.

**Transfer across architectures.** We also study how the prior contrastive adapting improvements transfer to other pretrained models. Table 4 shows contrastive adapters substantially improve group robustness for models such as CLOOB [20]. The method also scales across model sizes, raising worst-group accuracy by 33.7 to 61.7 pp via training adapters with only 0.52% to 1.03% of the model parameters [20, 59].

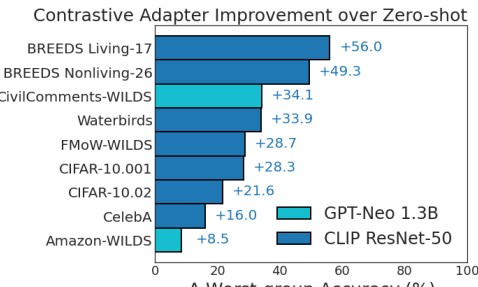

Contrastive Adapter Improvement over Zero-shot

Figure 4: Across 9 group robustness benchmarks, contrastive adapting consistently improves worst-group acc. over pretrained zero-shot classification.

### 5.2 Ablation on sampling strategy and contrastive training objective

To next better understand how contrastive adapting's individual components affect group robustness, we ablate the proposed contrastive objective (Eq. 6) and "hard" sampling strategy, and report worst-group and average accuracies on CLIP RN-50 adapters (Table 5). On three datasets, we find the contrastive loss alone improves robustness more than hard sampling alone. However, on Waterbirds and CelebA—where ERM adapters perform poorly—having both components substantially improves robustness (+5.5 to 8.5 pp). Meanwhile, on BREEDS Living-17 and CIFAR-10.02—where ERM adapters perform best across all methods—removing hard sampling improves contrastive adapting performance. On these datasets, the random sampling in ERM may be *beneficial* (discussed further in App. E.6). Contrastive adapting may thus also benefit from random sampling in these settings.

### 5.3 Measuring efficiency among effective group robustness solutions

While in Section 5.1, we found contrastive adapters could significantly improve group robustness for foundation models, we now expand on contrastive adapting's efficiency. We find that for group robust classification in general, contrastive adapting can achieve state-of-the-art performance despite only training ≤1% of the usual model parameters. The lightweight nature of contrastive adapting also leads to better data efficiency than existing state-of-the-art approaches.

Table 5: Contrastive adapter ablation over contrastive loss and sampling. When ERM obtains poor worst-group (WG) accuracy, both contrastive loss and "hard" sampling lead to best robustness. When ERM improves WG acc. over zero-shot (c.f. Table 3, *i.e.* random sampling helps), no hard sampling also helps contrastive adapters.

| Adapter Ablation | Contrastive Loss | Hard Sampling | Waterbirds | | CelebA | | BREEDS Living-17 | | CIFAR-10.02 | |
|---|---|---|---|---|---|---|---|---|---|---|
| | | | WG | Avg. | WG | Avg. | WG | Avg. | WG | Avg. |
| ERM | ✗ | ✗ | 60.8 ± 0.9 | 96.0 ± 0.1 | 36.1 ± 1.4 | 94.2 ± 0.2 | **70.7 ± 0.9** | 94.0 ± 0.1 | **68.8 ± 0.5** | 86.0 ± 0.5 |
| Hard sample only | ✗ | ✓ | 56.3 ± 1.5 | 81.4 ± 0.5 | 84.5 ± 3.2 | 92.6 ± 0.4 | 58.7 ± 4.9 | 89.6 ± 0.8 | 58.5 ± 2.0 | 80.4 ± 0.7 |
| Contrastive only | ✓ | ✗ | 75.2 ± 1.0 | 94.0 ± 0.1 | 51.4 ± 5.9 | 93.2 ± 2.6 | 67.4 ± 0.9 | 91.8 ± 0.2 | 66.9 ± 1.2 | 82.9 ± 0.3 |
| Default proposed | ✓ | ✓ | **83.7 ± 0.7** | 89.4 ± 0.9 | **90.0 ± 0.4** | 90.7 ± 0.4 | 62.0 ± 1.6 | 90.9 ± 0.3 | 60.7 ± 1.7 | 80.9 ± 0.2 |

Table 6: On popular Waterbirds and CelebA benchmarks, contrastive adapters achieve near state-of-the-art worst-group accuracy (WG Acc.) with ≤1% of the trainable parameters. ΔAcc. is percentage point gap with prior SoTA. **1st** / 2nd best metrics **bolded** / underlined. We report numbers from original works.

| Model | # Trained Params | % Params | Method | Waterbirds WG Acc. (%) | ΔAcc. | CelebA WG Acc. (%) | ΔAcc. |
|---|---|---|---|---|---|---|---|
| ResNet-50 | 25557032 | 100 | EIIL [16] | 78.7 | -10.3 | 83.3 | -6.5 |
| | | | CIM [72] | 83.6 | -5.4 | 83.6 | -6.2 |
| | | | JTT [47] | 86.7 | -2.3 | 81.1 | -8.7 |
| | | | RWY [34] | 86.1 | -2.9 | 82.9 | -6.9 |
| | | | CNC [79] | 88.5 | -0.5 | 88.8 | -1.0 |
| | | | SSA [52] | **89.0** | 0.0 | 89.8 | 0.0 |
| Adapter + CLIP RN-50 | 263424 | 1.03 | Ours | 83.7 | -5.3 | **90.0** | 0.2 |
| Adapter + CLIP ViT-L/14 | 197632 | 0.77 | Ours | 86.9 | -2.1 | 84.6 | -5.2 |

**Robustness comparison to state-of-the-art methods.** In Table 6, we evaluate how contrastive adapting with CLIP RN-50 and ViT-L/14 embeddings compares to current state-of-the-art robustness techniques. We use the popular Waterbirds and CelebA datasets. Existing group robustness methods train ImageNet-pretrained ResNet-50s. On both datasets, contrastive adapting achieves comparable worst-group accuracy to these methods, despite only training ≤1% of their parameters. Notably, contrastive adapting outperforms some methods by up to 5.0 and 10.1 pp for Waterbirds and CelebA, and only falls short of the state-of-the-art Spread Spurious Attribute (SSA) method by 2.1 pp on Waterbirds. These results suggest contrastive adapters not only effectively improve group robustness for pretrained models, but also enable competitive robust classification in general at a fraction of prior approaches' trainable parameter counts.

**Data efficiency evaluation.** Beyond model parameter count, we also study if the lightweight nature of contrastive adapting transfers to better data efficiency. We compare contrastive adapting to the best performing SSA on subsampled versions of Waterbirds. To evaluate how well methods maintain group robustness with less training data available, we keep group ratios preserved (*i.e.*, 1% of all training samples belongs to the smallest Waterbirds group [65]).

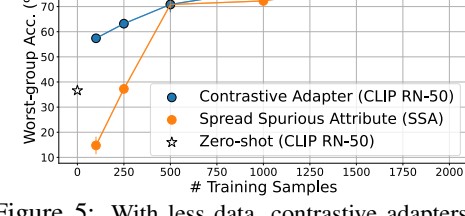

Figure 5: With less data, contrastive adapters maintain higher group robustness than zero-shot and significantly outperform standard models trained with the state-of-the-art SSA method.

Figure 5 shows that contrastive adapting substantially outperforms SSA in lower data regimes. With only 100 and 250 training samples, contrastive adapting outperforms SSA-trained ResNet-50s by 42.6 and 25.9 pp. With 100 training samples, contrastive adapting still achieves 20.8 pp higher worst-group accuracy than zero-shot classification with only 1 training sample in the smallest Waterbirds group. In contrast, SSA's accuracy drops significantly, resulting in 17.6 pp lower worst-group accuracy than zero-shot classification. To connect this fewer data result to our generalization discussion in Section 4.2, with prior methods [19] we estimate the trained adapter Lipschitz constant. We estimate the constant to be 29.3, much lower than that reported for larger networks (*e.g.*, ResNet-50s) [19, 73].

## 6 Conclusion

We study the group robustness of popular foundation models. We find their zero-shot classification may not be robust to various group shifts, establish that baseline linear probe and adapter strategies do not reliably improve robustness, and propose a simple adapter strategy to significantly and consistently improve FM robustness without finetuning. This suggests FM embeddings do contain group-relevant information, and we show that we can use FM embeddings to efficiently achieve state-of-the-art robust classification. We recognize the limitations of computational solutions to subgroup performance disparities, and the need to understand FMs in broader socio-technical systems [9].

# 7 Acknowledgements and Funding

We thank Simran Arora, Megan Leszczynski, Kush Bhatia, Maya Varma, Gautam Machiraju, and Laurel Orr for helpful discussions and feedback.

We gratefully acknowledge the funding support of NIH under No. U54EB020405 (Mobilize), NSF under Nos. CCF1763315 (Beyond Sparsity), CCF1563078 (Volume to Velocity), and 1937301 (RTML); ARL under No. W911NF-21-2-0251 (Interactive Human-AI Teaming); ONR under No. N000141712266 (Unifying Weak Supervision); ONR N00014-20-1-2480: Understanding and Applying Non-Euclidean Geometry in Machine Learning; N000142012275 (NEPTUNE); NXP, Xilinx, LETI-CEA, Intel, IBM, Microsoft, NEC, Toshiba, TSMC, ARM, Hitachi, BASF, Accenture, Ericsson, Qualcomm, Analog Devices, Google Cloud, Salesforce, Total, the HAI-GCP Cloud Credits for Research program, the Stanford Data Science Initiative (SDSI), and members of the Stanford DAWN project: Facebook, Google, and VMWare. The U.S. Government is authorized to reproduce and distribute reprints for Governmental purposes notwithstanding any copyright notation thereon. Any opinions, findings, and conclusions or recommendations expressed in this material are those of the authors and do not necessarily reflect the views, policies, or endorsements, either expressed or implied, of NIH, ONR, or the U.S. Government.

We have no other additional revenues to disclose related to this work.

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
