# OpenReview forum: "Contrastive Adapters for Foundation Model Group Robustness"
_NeurIPS.cc/2022/Conference — NeurIPS 2022 Accept_

### Official Review · Reviewer_8jHt · 2022-07-07

**Rating:** 7
**Confidence:** 4
**Soundness:** 3 good
**Presentation:** 3 good
**Contribution:** 3 good

**Summary:**

This work proposes a new method to improve group (subpopulation) robustness in foundational models like transformers. The problem of group robustness (in terms of worst-group performance) is particularly challenging when some groups are underrepresented in the training data. Classical approaches use under/over-sampling, reweighting, and two-phase training to deal with this issue. The authors propose to use contrastive adapters without retraining from scratch. The idea is to contrast sample embeddings from the same class to improve the embedding structure. The paper provides experimental evidence that such an approach is effective when dealing with different kinds of shifts in the data, obtaining good performance.

**Questions:**

* Regarding the contrastive adapter name: isn't the (standard) adapter contrastive too if we follow your argument in 4.1 and Eq. 5?

* In Table 2) and Table 3) is the (standard) adapter trained by ERM or using the InfoNCE loss in Eq.5?
    * If you use ERM, I would like to see the adapter performance trained explicitly with Eq 5.
    * Additionally, it would be interesting to see the adapter performance training with Eq 5 only, Eq 6 only, and both equations.

* How large is P for Eq.6? Did you try to use different values for P?

* How do you mix CIFAR10 and CIFAR02?

**Limitations:**

* The method seems to work well only with a few classes (binary classification), and performance deteriorates for different shifts and more complex datasets like CIFAR10.02. In particular, increasing the number of classes, the standard adapter and the linear probe work better than the contrastive adapter (BREEDS and CIFAR in Table 3). This is not the behavior I would expect if the argument in 4.1 is correct.

* Eq 5 and Eq 6 are both contrastive losses. You link the ERM adapter with the InfoNCE but did not use Eq 5 to train the standard adapter (as far as I understand). So this connection is not backed by experiments.

* Looking at Table 2, I agree that all the baselines fail to improve robustness consistently. But so your method in Table 3.

* I would compare your approach with JM1 (https://openreview.net/forum?id=HI2ilxFli0W): both methods have access to fine-grained label structure. The "retraining" in JM1 starts from a pre-trained foundational model.

**Strengths And Weaknesses:**

strengths

* The paper is well written and easy to follow with a clear goal (improving group robustness in foundational models).
* The problem tackled is relevant for the use and deployment of foundational models.
* The idea of "contrastive adapter" is novel.
* Method easy to implement and to apply to a large variety of scenarios.

weaknesses

* The idea of increasing the adapter contrasting not only class and sample but also samples from the same class is incremental.
* Results do not clearly show that contrastive adapters work better (in terms of robustness) with all the considered distribution shifts in Table 3. In particular, standard adapters seem to work better when increasing the dataset complexity (number of classes in CIFAR10).

---

> ### Author Response · Authors · 2022-08-02
> **Response to Reviewer 8jHt (1/3)**
>
> Thank you for your review! We appreciate that you found our problem relevant and our method effective and easy to implement and apply in various settings. In our response, we clarify our evaluation aims, comparison to baselines, and details on how to interpret how the method we proposed performs in different settings.
>
> We believe that after clarification, some of your questions are actually already addressed in the original submission. We have updated the revision to improve the presentation. Please find our detailed comments to your questions and concerns below.
>
> **Adding samples to contrast during adapter training is incremental**
> While a simple extension, we make this connection with standard adapter training to point out an unexploited way to improve performance. We also note that part of our contribution here is our “hard sampling” method for contrastive adapter training (L239-244). We make this more clear in the revision with an ablation of the sampling strategy (App E.2.2). Merely adding samples alone to contrast (e.g. as in supervised contrastive learning [34]) does not improve robustness as much as our proposed approach on datasets such as Waterbirds and CelebA.
>
> Furthermore, we reiterate that the contrastive adapter method is just one of the contributions in the paper. We also:
> * Present the group robustness issue for foundation models and show that existing baselines do not consistently improve robustness (Table 1).
> * Find that despite this initial poor performance on certain groups, popular foundation models often *do* encode the sufficient information to classify these groups, evidenced by how we can significantly improve group robustness by training on top of these embeddings (Table 3).
> * Find that the combination of pretrained embeddings + our proposed adapter strategy can obtain state-of-the-art group robustness on popular benchmarks, while only training a fraction of the parameters of other approaches (Table 5).
>
> **Evaluation, improvement over standard adapters, consistent improvement over zero-shot baseline**
> Regarding our evaluation, we clarify that our aim is not to outperform standard ERM-trained adapters in all scenarios. Rather, we aim to solve the problem that existing approaches (like standard adapter training via ERM) *do not consistently* improve robustness over the baseline zero-shot classification for foundation models (L49-53, L58) (Table 1).
>
> Table 3 shows our proposed approach *does consistently* improve robustness over zero-shot classification, satisfying this aim. Furthermore, while our approach does not perform better than every other method everywhere, Table 3 also shows that on average, our method achieves 12.4 and 4.1 pp higher worst-group accuracy than the next best method with CLIP RN-50 and CLIP ViT-L/14 models.

---

> > ### Author Response · Authors · 2022-08-02
> > **Response to Reviewer 8jHt (2/3)**
> >
> > **“The method seems to work well only with a few classes (binary classification), and performance deteriorates for different shifts and more complex datasets. In particular, increasing the number of classes, the standard adapter and the linear probe work better than the contrastive adapter (BREEDS and CIFAR)”**
> >
> > We respectfully believe this is a slight mischaracterization of our method. The BREEDS and CIFAR-10 datasets, rather than being more complex (with regard to achieving better group robustness), carry properties that make ERM more competitive. Our current hypothesis is that this is because BREEDS Liv-17 and CIFAR-10.02 have balanced groups, and less distribution shifts between groups, respectively. We note that our method still consistently and significantly improves robustness over the zero-shot pretrained model (Table 3).
> >
> > We support this reasoning with additional experiments in App. E.6. We discuss the effects of balanced groups and BREEDS Liv-17 in our response here; please see our revision for similar clarifications for CIFAR-10.02 (App. E.6.1).
> >
> > To better understand if the number of classes matters, and/or if balanced groups play a factor, we evaluate the ERM linear probe, ERM adapter, and contrastive adapter on a *harder* version of the BREEDS Liv-17 dataset (w.r.t. group-robust classification) by downsampling certain groups in the training data.
> >
> > Note that the BREEDS Liv-17 dataset has 17 classes, and by default groups in each class are roughly equally sized (Table 2). We can make group robust classification harder by introducing group imbalance, as models may only learn correlations that hold for classifying the majority groups (and get poor worst group performance) [58]. When we instead downsample the training data such that for each class, one group only makes up only 5% of the class (as in Waterbirds), we find that contrastive adapters substantially outperform the ERM adapters and linear probes (c.f. Table 19 in revision, reproduced below, comparison horizontally by row).
> >
> > | CLIP RN-50 WG Acc. (%) | ERM Linear Probe |   ERM Adapter  | Contrastive Adapter |
> > |------------------------|:----------------:|:--------------:|:-------------------:|
> > | Group Balanced (default)         |    53.3 ± 0.9    | **70.7 ± 0.9** |      62.0 ± 1.6     |
> > | Group Imbalanced       |     8.0 ± 0.0    |   56.0 ± 1.6   |    **60.0 ± 2.3**   |
> >
> > | CLIP ViT-L/14 WG Acc. (%) | ERM Linear Probe | ERM Adapter | Contrastive Adapter |
> > |---------------------------|:----------------:|:-----------:|:-------------------:|
> > | Group Balanced (default)            |  **84.0 ± 0.9**  | 82.8 ± 0.9 |      80.0 ± 1.6     |
> > | Group Imbalanced          |    52.7 ± 1.9    |  70.7 ± 0.9 |    **72.7 ± 1.9**   |
> >
> > Thus even in multiclass classification and different group-shift settings, contrastive adapters can outperform ERM alternatives. In particular, our results suggest that our method is more desirable when group-robust classification is arguably harder due to imbalanced groups in the training data. We began characterizing this trade-off between data setting and method in L186 - 190, and expand upon this in our revision (App. E.6.2).
> >
> > We also note that in practice, we may not know the training data characteristics ahead of time. Due to the efficiency of training adapters, practitioners could train multiple adapters (e.g., one via ERM and one via our approach), and select the best through validation. We found that ERM adapters may not always suffice for improving robustness over zero-shot classification with large pretrained models (Table 1). Thus we believe our contribution is still meaningful to expand the scenarios where we can efficiently improve group robustness without fine-tuning these large petrained models.
> >
> > **Isn’t the standard adapter “contrastive” too?**
> > Good point! Standard adapter training with prompt embeddings is also interpretable as contrastive learning (CL). However, because standard adapter training only makes use of prompt embeddings as positives and negatives, it is only a limited CL (which typically benefits from large numbers of positives and negatives [13]).
> >
> > Our proposed approach better captures the benefits of CL by allowing us to sample large numbers of positives and negatives from other sample embeddings.

---

> > > ### Author Response · Authors · 2022-08-02
> > > **Response to Reviewer 8jHt (3/3)**
> > >
> > > **Clarification of standard adapter training, ERM, and connection to InfoNCE**
> > > We clarify that the standard adapter is trained *via* ERM to minimize the sample InfoNCE loss (Eq. 5). This is actually standard practice; our connection just follows from how the sample InfoNCE loss in the adapter & zero-shot pretrained model setting [54, 21] is the same as the sample cross-entropy loss used in standard classification.
> > >
> > > Recall that for cross-entropy, we need class probabilities, i.e. for correct class index $i$,
> > > $
> > > \mathcal{L}_{\text{ce}} = - \log p_i
> > > $.
> > > As detailed in (L148-152, 167-168) and prior work [3, 4], we get probability $p_i$ for a given sample and a set of $C$ classes by:
> > > 1. Computing the distances between the sample embedding (adapter output or pretrained embedding) and each pretrained class embedding (via cosine similarity, or dot product if the embeddings are normalized)
> > > 2. Taking the softmax over the distances (with some temperature weighting)
> > >
> > > So if $f_\theta(u)$ is the adapter-outputted sample embedding, $v_i$ is the ground-truth class embedding, $\\{v_c\\}\_{c=1}^C$ is the set of all class embeddings, and $\tau$ is the temperature, then
> > >
> > > $p_i = \frac{ \exp(\cos(f_\theta(u), v_i) / \tau) }{ \sum_{c = 1}^C \exp (\cos(f_\theta(u), v_c) / \tau) } $
> > >
> > > Plugging $p_i$ back into the cross-entropy loss, we get Eq. 5, equating the cross-entropy loss and InfoNCE loss provided.
> > >
> > > We clarify this in our revision. In particular, we note that when training with ERM, we are also training to minimize Eq. 5 (L214-215). This connection is again backed by the walkthrough above. For consistency, Eq. 3 in the first draft should not have the softmax, and we should refer to the outputs as logits not probabilities. Apologies for this error; we fixed this in the revision.
> > >
> > >
> > > **Adapter ablation over objectives in Eq. 5, Eq. 6, and both**
> > > We included this ablation in Table 13 of our original full paper submission. The default contrastive adapter is trained with both equations, as noted in L246 - 247. L246 - 250 also includes our reasoning for this, as we want to update the sample embeddings with the adapters but still have them remain “close” to their ground-truth prompt embeddings. Our results support this reasoning on Waterbirds: when removing either component, we see drops in group robustness. Without Eq. 6 (the contrastive loss), we miss out on training with additional contrastive samples. However, without Eq. 5 (the cross-entropy loss), we do not guarantee that the adapter outputs are close to their ground-truth class embeddings.
> > >
> > > **How large is P for Eq.6? Did you try to use different values for P?**
> > > The number of positives (P) varies per dataset (either 512 or 2048, c.f. Table 9 in the full paper). We tried different values for P by treating P as a hyperparameter (L728 - 731, App. C.1). We also explored the relation between number of positives and group robustness more extensively in App. E.2.2. (E.2.3 in the revision). One benefit of our method is that as we train with pretrained embedding inputs (e.g. size 1 x 1024), we can train with very large P with very little GPU memory; prior work shows large P is beneficial for contrastive learning [13].
> > >
> > > **How do you mix CIFAR-10 and CIFAR-10.2?**
> > > For each class (shared by both datasets), we combine samples from CIFAR-10 and CIFAR-10.2 such that ~10% of every class in the combined dataset comes from CIFAR-10.2. We include additional details on the dataset construction in the revision (App. A.1, L651 - 654).
> > >
> > > **Comparison to JM1 [23]**
> > > We added this comparison in App. E.5 in the revision, and report comparisons for CLIP RN-50 pretrained embeddings below. On three of the four main evaluation datasets, the contrastive adapter approach achieves higher worst-group (WG) accuracy than JM1 (reported from Table 17):
> > >
> > > | Acc. (%)                               | Waterbirds WG | Waterbirds Avg | CelebA WG  | CelebA Avg | BREEDS Liv-17 WG | BREEDS Liv-17 Avg | CIFAR-10.02 WG | CIFAR-10.02 Avg |
> > > |----------------------------------------|---------------|----------------|------------|------------|------------------|-------------------|----------------|-----------------|
> > > | JM1      |   74.2 ± 3.1   | 80.4 ± 1.0 |   87.1 ± 1.1   | 91.6 ± 1.0 |   61.7 ± 2.0   | 91.1 ± 0.1 | **65.2 ± 0.9** | 82.6 ± 0.2 |
> > > | Ours     | **83.7 ± 0.7** | 89.4 ± 0.9 | **90.0 ± 0.4** | 90.7 ± 0.4 | **62.0 ± 1.6** | 90.9 ± 0.3 |   60.7 ± 1.7   | 80.9 ± 0.2 |
> > >
> > > **References**
> > > [13] Chen et al. A Simple Framework for Contrastive Learning of Visual Representations. 2020
> > > [21] Gao et al. Clip-Adapter: Better Vision-Language Models with Feature Adapters. 2021
> > > [23] Giannone et al. Just Mix Once: Mixing Samples with Implicit Group Distribution. 2021
> > > [34] Khosla et al. Supervised Contrastive Learning. 2020
> > > [54] Radford et al. Learning Transferable Visual Models from Natural Language Supervision. 2021
> > > [58] Sagawa et al. Distributionally Robust Neural Networks for Group Shifts. 2019

---

> > > > ### Comment · Reviewer_8jHt · 2022-08-05
> > > > **Comments**
> > > >
> > > > Thanks for the rebuttal. I appreciate the explanations and additional ablation studies the authors performed. I raised my score.
> > > > Regarding the title: given that the standard adapters are contrastive too (even if in a simplified setting), it could make sense to modify the paper title using "Multi-view Contrastive Adapters..." or similar. I think the title can be confusing without previous knowledge of the subfield.

---

> > > > > ### Author Response · Authors · 2022-08-07
> > > > > **Follow-up**
> > > > >
> > > > > Thank you again for your review! We appreciated the insightful questions, which helped us clarify and provide new understanding in the paper. We also very much appreciate you raising your score.
> > > > >
> > > > > Regarding a title change, we are looking into this for the final paper. We agree that a modifier such as “Multi-view” or “Sample-embedding” would be helpful for clarity. Based on last year’s FAQs, we may need to email the program chairs to make the change: https://neurips.cc/Conferences/2021/PaperInformation/NeurIPS-FAQ/#:~:text=Can%20I%20change%20the%20title%20of%20my%20paper%3F

---

### Official Review · Reviewer_TAXS · 2022-07-10

**Rating:** 7
**Confidence:** 5
**Soundness:** 3 good
**Presentation:** 3 good
**Contribution:** 3 good

**Summary:**

This paper studies the group robustness of foundation models, and shows that large scale pretrained foundation models can still have a bad robustness on some group shifts. Then the paper proposed a contrastive adapting method that take advantage of supervised contrastive learning to train the adapter to improve the alignment and cosine similarity metrics. Experimental results show that the proposed contrastive adapter is successful in improving the group robustness of the foundation models.



**Questions:**


Q1: In fig 3, we can see that only on CelebA that the learned adapter has a worse group robustness, but as in tab 2, the group shift in CelebA and Waterbirds are the same (Confounder), why would the robustness of learned adapter on waterbirds be better than zero shot while on CelebA it's the other way around? Is something other than the group shift influencing the robustness of the model?

Q2: It is known that supervised contrastive learning (SupCon) outperforms regular supervised training with CE loss. It would be good if the paper could provide an analysis of how does SupCon helps the group robustness of foundation models, could it be that the performance gain is because SupCon improves the performance of the model overall (including improvement in IID performance), and the improvement in group robustness is a side effect?

Q3: If I understood correctly, the proposed solution only tunes the adapter on the feature from the image encoder of the CLIP model, I am wondering how this method compare to the visual-language adapters like in [R1] or [R2].

Q4: The proposed method is quite similar to the ones proposed in [57], both uses supervised contrastive to help the group robustness, I would like to see a comparison between the proposed method in this paper and the one in [57].


[R1] Learning to Prompt for Vision-Language Models, arXiv 2021.

[R2] Conditional Prompt Learning for Vision-Language Models, CVPR 2022.


**Limitations:**


1. This work proposed a way to improve the foundation model's performance wrt to group robustness, however, the method still needs to run inference on the entire foundation model which can be computational expensive.


**Strengths And Weaknesses:**


S1: The problem this paper tries to tackle is practically important, as the foundation model is used in more applications, the robustness of these models is important to study.

S2: The performance of proposed method seems to be improving over the baseline by a large margin.

W1: Some of the experimental results are not very clearly explained, see Questions.

W2: The supplementary material is not properly anonymized, the `.git` folder in the code submission contains a commit message that have the author's name on it.


Note that most of this review is completed before I unwillingly find out the identity of the author.

---

> ### Author Response · Authors · 2022-08-02
> **Response to Reviewer TAXS (1/2)**
>
> Thank you for your review! We appreciate that you noted the importance of our problem and the effectiveness of our proposed method. We updated our revision with several suggested comparisons and clarifications to address your comments. We answer your individual questions below.
>
> **Q1: Why different Waterbirds vs CelebA adapter results?**
> While both datasets exhibit spurious confounders, there are likely unexplained differences in the available training data. Note that training on both these datasets can lead to poor worst-group accuracy, as models can learn spurious correlations between the confounder and the ground-truth class that only hold for some groups (L136 -138). This can further bias classification and lead to worse group robustness than zero-shot, as we see with linear probes on both datasets in Table 1.
>
> Even with a more powerful model (e.g. an adapter), the spurious correlations in CelebA may be “harder” to overcome. This is evidenced by prior robustness work [42, 58, 64] that also shows ERM-trained ResNet-50s achieve higher worst-group accuracy after training on Waterbirds than CelebA.
>
> However, a definitive answer is hard to give, as the relation between pretraining data (for the embeddings), downstream task data, and downstream task performance for these large models is not well-characterized. We think this question, and understanding these relations, point to an exciting and important direction for future work!
>
> **Q2: Analysis on how SupCon helps improve group robustness vs overall performance**
> Thanks for this suggestion. In our revision (App. E.2.2), we conducted such an analysis, finding that while supervised contrastive learning can lead to higher average accuracy and group robustness, by itself it does not always lead to most improved group robustness. Rather, for datasets where standard training can fail to improve group robustness (Table 1), our proposed “hard sampling” procedure (L239 - L244) plays a key role in substantially improving robustness (see table below, e.g. +38.6pp worst-group (WG) accuracy on CelebA with the hard sampling).
>
> For multiple datasets, we compare training adapters with our proposed approach (hard sampling + contrastive loss), supervised contrastive loss without the sampling (i.e. as in vanilla SupCon where positives / negatives are just picked by having same / different classes), and the cross-entropy loss. We report results from Table 14 below.
>
> | Accuracy (%)                            | Waterbirds WG  | Waterbirds Avg | CelebA WG      | CelebA Avg     |
> |-----------------------------------------|:--------------:|:--------------:|:--------------:|:--------------:|
> | Cross-entropy / ERM                     | 60.8 ± 0.9     | 96.0 ± 0.1     | 36.1 ± 1.4     | 94.2 ± 0.2     |
> | No Hard Sampling + Contrastive (SupCon) | 75.2 ± 1.0     | **94.0 ± 0.1** | 51.4 ± 5.9     | **93.2 ± 2.6** |
> | Hard Sampling + Contrastive (Ours)      | **83.7 ± 0.7** | 89.4 ± 0.9     | **90.0 ± 0.4** | 90.7 ± 0.4     |
>
> **Q3: Comparison to CoOp and CoCoOp**
> We tried running CoCoOp, which extends and outperforms CoOp as a follow-up method [75], using the official provided code: https://github.com/KaiyangZhou/CoOp
>
> Unfortunately, despite an ample hyperparameter sweep (lr: 1e-3, 2e-3, 5e-5; weight decay 1e-1, 1e-3, 1e-4, 1e-5), we have not yet been able to get competitive performance. Our best result was 82.5% worst-group accuracy on CIFAR-10.02 training with CLIP ViT-L/14, but we only managed to get 28.9 ± 2.3% worst-group acc. (57.3% average acc) on Waterbirds (training with early stopping over 300 epochs).
>
> We initially did not include comparison to CoCoOp (or CoOp) due to several key differences and downsides. First, the methods require additional assumptions. As CoCoOp optimizes prompts as *inputs* to the pretrained model, CoCoOp requires either access to model weights or (expensive) iterative queries to an API for every sample and epoch during training. These are avoided with our approach, because we just train on top of frozen pretrained model outputs (the embeddings). CoCoOp is also more expensive to train. For CLIP, CoCoOp requires forward and backward passes through the entire CLIP model and processing input images (e.g. tensor shape 3 x 224 x 224). Instead, our approach only trains on top of pretrained embeddings (e.g. tensor shape 1 x 1024). Thus we only compute embeddings for samples once, and process much smaller inputs (~0.7% of the size). This amounts to much faster training with our adapter approach than CoCoOp: on Waterbirds we reach higher worst-group accuracy in 12 minutes than CoCoOp achieves in 1.3 hours.

---

> > ### Author Response · Authors · 2022-08-02
> > **Response to Reviewer TAXS (2/2)**
> >
> > **Q4: Comparison to CNC**
> > In Table 5 of our original submission, we compared against the reported results of CNC from the paper. While not the same setup (CNC trains two ResNet-50s, we use the zero-shot predictions of a pretrained model and only train an adapter), the contrastive adapter on pretrained CLIP ResNet-50 embeddings gets comparable worst-group accuracy (-4.8pp on Waterbirds, +2.0pp on CelebA), while only training 1% of the parameters.
> >
> > In our revision, we also added a closer comparison to CNC implemented for our setting, training a robust adapter (MLP) on top of pretrained model embeddings (Table 17, App. E.5). We adopt CNC’s 2-stage procedure: (1) train an adapter with ERM to predict classes, (2) train a robust adapter with CNC’s robust sampling and contrastive objective using these predictions. Results are reported below:
> >
> > | Acc. (%)                               | Waterbirds WG | Waterbirds Avg | CelebA WG  | CelebA Avg | BREEDS Liv-17 WG | BREEDS Liv-17 Avg | CIFAR-10.02 WG | CIFAR-10.02 Avg |
> > |----------------------------------------|---------------|----------------|------------|------------|------------------|-------------------|----------------|-----------------|
> > | CNC     |   82.7 ± 2.0   | 87.0 ± 1.1 |   86.9 ± 1.4   | 91.8 ± 0.4 |   60.7 ± 2.5   | 88.1 ± 0.7 |   59.2 ± 2.5   | 80.4 ± 0.2 |
> > | Ours    | **83.7 ± 0.7** | 89.4 ± 0.9 | **90.0 ± 0.4** | 90.7 ± 0.4 | **62.0 ± 1.6** | 90.9 ± 0.3 | **60.7 ± 1.7** | 80.9 ± 0.2 |
> >
> > While similar in objective, we find slightly higher worst-group (WG) accuracy with the contrastive adapter (Ours). We reason this is because in contrastive adapting we specifically sample negatives from the nearest neighbor samples to anchor in a different class (L243). Focusing on these harder anchor-negative comparisons may lead to learning representations better suited for group robust classification (also see App. E.5.3 for additional discussion).
> >
> > **Inference Limitation**
> > We note that running inference with the foundation model is necessary for *all* methods that hope to use these large pretrained models' outputs. While potentially costly, as discussed above, we can reduce this cost substantially compared to other methods by only running inference once for each sample, saving their outputs (here the frozen embeddings), and only training small MLPs on top of these frozen embeddings. We do not require any additional inference with the pretrained model at any other time during training.
> >
> > **Code**
> > We sincerely apologize for accidentally including the .git folder. We have removed it in our revised supplement.
> >
> > **References**
> > [42] Liu et al. Just Train Twice: Improving Group Robustness Without Training Group Information. ICML 2021.
> > [58] Sagawa et al. Distributionally Robust Neural Networks for Group Shifts. ICLR 2019.
> > [64] Sohoni et al. No Subclass Left Behind: Fined-Grained Robustness in Coarse-Grained Classification Problems. NeurIPS 2020.
> > [75] Zhou et al. Conditional Prompt Learning for Vision-Language Models. CVPR 2022.

---

> > > ### Comment · Reviewer_TAXS · 2022-08-04
> > > **Thanks for the rebuttal**
> > >
> > > Q1: Thanks for the clarification. I agree that a clear answer can not be given right now, and may be out of the scope of this paper.
> > >
> > > Q2: Thanks for the additional results, the results are clear and I think it makes sense to me, I would suggest including this in the main paper in a future version as it is one of the most important baselines to consider.
> > >
> > > Q3: Thanks for the additional results, I think the argument is convincing.
> > >
> > > Q4: This is clear to me now, thanks.
> > >
> > > Additional suggestion:
> > >
> > > Some works should also be cited as they study OOD robustness which is relevant to this paper.
> > >
> > > [1] Benchmarking Neural Network Robustness to Common Corruptions and Perturbations, ICLR 2019
> > > [2] Natural Adversarial Examples, CVPR 2021
> > > [3] ROBIN: A Benchmark for Robustness to Individual Nuisances in Real-World Out-of-Distribution Shifts, ECCV 2022
> > >
> > > I would suggest including a table of content for the supplementary as it is hard to navigate for now.
> > > I would also suggest using the term 'large pretrained model' rather than 'foundational model' in a future version of the paper, for the same reason as Reviewer 2hnT

---

> > > > ### Author Response · Authors · 2022-08-07
> > > > **Follow-up to Reviewer TAXS**
> > > >
> > > > Thanks again for your review! We are very happy to hear that we could address all of your questions. We appreciate the additional feedback; we followed your suggestions in the latest uploaded revision to improve our paper’s presentation:
> > > > 1. We moved the ablation of contrastive adapter components (contrastive objective, sampling strategy baselines) to the main paper (Section 5.2)
> > > > 2. We added the suggested citations on OOD robustness ([31, 32, 82])
> > > > 3. We added a table of contents for the appendix (page 17)
> > > >
> > > > We are also working on changing instances of the term “foundation model” to “large pretrained model” for the final paper. Right now we are a bit pressed with space, but hope the extra page in the final paper can help.
> > > >
> > > > We are more than happy to help clarify any other standing questions or concerns. Given that we addressed all of the questions in your initial review, we kindly ask if you would be willing to adjust your review score while taking our rebuttal into account.
> > > >
> > > > **References**
> > > > [31] Hendrycks & Dietterich. Benchmarking Neural Network Robustness to Common Corruptions and Perturbations. ICLR 2019.
> > > > [32] Hendrycks et al. Natural Adversarial Examples. CVPR 2021.
> > > > [82] Zhao et al. ROBIN: A Benchmark for Robustness to Individual Nuisances in Real-World Out-of-Distribution Shifts. ECCV 2022.

---

> > > > > ### Comment · Reviewer_TAXS · 2022-08-07
> > > > > **Thanks**
> > > > >
> > > > > Thank you for your new updates, I have no concerns with the paper now, and I have raised my score to 7

---

### Official Review · Reviewer_2hnT · 2022-07-11

**Rating:** 6
**Confidence:** 4
**Soundness:** 3 good
**Presentation:** 2 fair
**Contribution:** 3 good

**Summary:**

The paper studies the classification group robustness of large-scale image-text pre-trained models. The authors identified that, under the prompt-based zero-shot classification protocol, the worse-group classification accuracy is considerably lower than the average group classification accuracy. The authors proposed to use a metric-learning loss (i.e., contrastive loss), on top of a MLP adapter from previous works, for finetuning on target datasets. Experimental results on a few benchmarks show improvement over baselines on worse-category classification accuracy.

**Questions:**

In the author response phase I encourage the authors to respond to the suggested baselines in my comments, with evidence of why they might be inferior or unnecessary.

**Strengths And Weaknesses:**

The paper studies a few interesting questions regarding these large-scale pre-trained models: 1) how to adapt these models for a target dataset, and 2) how do these models actually transfer across datasets and groups. These questions are important and they need to be addressed because the practice, and the hope of the community, is to rely more and more on the pre-trained models. The paper shows that neither a naive version of the zero-shot, nor linear probing on the target dataset, is the optimal choice. Benchmarked on standard datasets, The proposed method is effective across model architectures.

On the other hand, I am curious about a few baselines that I feel essential in evaluating the pre-trained models but seem to be missing from the draft. The first one is more prompt-tuning for zero-shot classification. Although impressed by the CLIP zero-shot classification accuracy, one must recognize the necessity of using properly tuned prompts or hints, as pointed out by the authors of the CLIP paper. It can be argued for the downstream evaluation/benchmarks in this paper such as WaterBirds, we can use prompts like "A land bird, on water background" to give more hints to the base model. I didn't find a discussion on how the zero-shot baseline is constructed or if meticulous tuning was conducted, and I believe this procedure is necessary for reliable conclusions.

Secondly, the metric-learning term (i.e., the contrastive loss) that is at the center of the proposed method looks heavily inspired by the hard-negative sampling strategy (see Line 235), acknowledged by the authors as well in Line 227. A potential ablation study on decoupling this hard-negative sampling procedure from the more complicated contrastive loss, in my opinion, is essential for understanding if the sampling procedure and/or the contrastive term is (more) important in the proposed method. If it is the sampling procedure that works, we don't necessarily need the embedding-based loss and it would be simple.

Thirdly, I can imagine a simple two-stage baseline that has the same essence of negative-sampling procedure. In the first stage, one can train a classifier using the naive sampling strategy. In the second stage, a new classifier can focus more on the misclassified samples, like a cascade classification framework. With a large-scale pre-trained model such as the CLIP model, the first stage can be its native zero-shot classifier. If interpreted this way, the sampling strategy proposed by the authors is is akin to the two-stage framework. Upon searching, a paper named *Just Train Twice: Improving Group Robustness without Training Group Information* does this for group robustness and it seems to be a valid baseline that should be studied in the paper.

Lastly, I'd like to raise my concern for using the term "foundation model" in an official and scientific research conference, despite it didn't play any role in my judgement for reviewing the submission. The practice of grouping a set of large-scale pre-trained models, large-scale being the amount of data, and/or the number of parameters in the model, and terming them "foundation" is subjective and without scientific background. Personally, I work on scaling models as well, and I'm a firm believer in the potential of these big models for the future of AI and ML research. It is precisely that the first-hand experience that makes me realize the flaws in this term. These models aren't foundational yet. They make silly mistakes all the time. They don't do grounding well. And they are far from reaching the ideal performance that we hope to get eventually, as evident in the experimental results of the submission as well. "Foundation model" is a hope, an inspiration, for now, not a reality. And a research paper is, by its nature, about the presence. Therefore in my humble opinion discarding the term will make the paper a stronger submission.

---

> ### Author Response · Authors · 2022-08-02
> **Response to Reviewer 2hnT (1/2)**
>
> Thank you for your review! We appreciated that you found the questions we tackled interesting and important. We also agreed with your comments on suggested baselines and comparisons; some of them were addressed in our original full paper submission (main paper + appendix). We apologize for the lack of clarity. Please find our clarifications below and in our revision. We also updated the paper to include additional suggested comparisons.
>
> **Prompt engineering for zero-shot baseline**
> In Appendix A.2, L668 - 670 of our original full paper submission (L681-684 in revision), we discussed that following Radford et al. [1], for each dataset we engineered prompts by trying several prompt templates. The reported zero-shot accuracy is from the template with the best validation worst-group accuracy. In our revision, we include all templates tried (Table 20), and clarify in the main paper that these details are in Appendix A.2 (L155-156).
>
> ***Using prompts with more hints***
> We included this baseline in our main paper submission. In L282-284 (285-287 in the revision), we discussed comparing against a “group-informed prompting” procedure, which uses prompts as suggested such as “A land bird on a water background". In our revision we clarify that this is referred to as Group Prompt ZS in the main results (Table 3).
>
> **Ablation to decouple hard-negatives vs contrastive loss**
> In Table 13 and App. E.2.1 of our original full paper submission, we ran a similar ablation. Using the same “hard” sampling procedure, we compared training adapters:
> * Without the contrastive loss.
> * Without the cross-entropy loss.
> * Using both losses (default).
>
> No contrastive loss still improved robustness over zero-shot, but having both losses was key for best performance (Table 13).
>
> In our revision (Table 14), we extended this ablation to the 4 datasets in our main results on CLIP RN-50. Following your suggestion, we compared:
> * Sampling hard negatives without the contrastive loss (only training via standard cross-entropy loss [same as (a.) above])
> * Using the contrastive loss, but without hard sampling (akin to standard supervised contrastive learning, i.e. SupCon [2])
>
> Comparing just contrastive vs. just hard sampling, we found just contrastive tended to obtain better robustness. Interestingly, we found that on datasets where standard ERM training (i.e. over random samples) performed best (e.g. BREEDS Living17, CIFAR-10.02; c.f. Table 3), removing the hard sampling component also improved contrastive adapter performance.
>
> However, we believe the hard sampling strategy still has merit: on other datasets where ERM training sometimes failed to improve robustness over zero-shot (Waterbirds, CelebA; c.f. Table 1), having both contrastive objective and hard sampling achieved best group robustness by a large margin.
>
> | Acc. (%)                               | Waterbirds WG | Waterbirds Avg | CelebA WG  | CelebA Avg | BREEDS Liv-17 WG | BREEDS Liv-17 Avg | CIFAR-10.02 WG | CIFAR-10.02 Avg |
> |----------------------------------------|---------------|----------------|------------|------------|------------------|-------------------|----------------|-----------------|
> | No contrastive loss; yes hard sampling | 56.3 ± 1.5    | 81.4 ± 0.5     | 84.5 ± 3.2 | 92.6 ± 0.4 | 58.7 ± 4.9       | 89.6 ±  0.8       | 58.5 ± 2.0     | 80.4 ± 0.7      |
> | Yes contrastive loss; no hard sampling | 75.2 ± 1.0    | 94.0 ± 0.1     | 51.4 ± 5.9 | 93.2 ± 2.6 | **67.4 ± 0.9**       | 91.8 ± 0.2        | **66.9 ± 1.2**     | 82.9 ± 0.3      |
> | Default    (both components)                            | **83.7 ± 0.7**    | 89.4 ± 0.9     | **90.0 ± 0.4** | 90.7 ± 0.4 | 62.0 ± 1.6       | 90.9 ± 0.3        | 60.7 ± 1.7     | 80.9 ± 0.2      |

---

> > ### Author Response · Authors · 2022-08-02
> > **Response to Reviewer 2hnT (2/2)**
> >
> > **Comparison to JTT**
> > Thanks for the connections here! We agree with them: in the main paper, we see our comparison to Deep Feature Reweighting (DFR) [3] as an analog to JTT, but better suited for adapting large pretrained models (L281 - 284).
> >
> > To see the similarities, note that both DFR and JTT train a more robust model after resampling the training data to balance groups. Because in our problem setting we don’t assume training group labels (L115), for DFR we first inferred these groups just like in JTT, using the zero-shot pretrained model predictions as the Stage 1 predictions (L284). However, recall that JTT as proposed would retrain an entire model (e.g. a new CLIP), which can be increasingly prohibitive as these models and their pretraining data get larger. Instead, DFR proposed to train a more robust *linear classifier* on a resampled set of a model’s last hidden-layer representations. This readily applies to our setting, where we can train such a classifier on pretrained model embeddings.
> >
> > Thus, for our setting, we saw DFR as a more applicable implementation of similar JTT resampling ideas. DFR was also previously proposed, so it better fit our scope of comparing *existing methods* to improve group robustness in the large pretrained model setting (i.e. no training of nor access to original model), and showing that adapters *could* consistently improve group robustness (via our proposed approach).
> >
> > Building on this, we think a comprehensive evaluation of how existing robust training methods transfer specifically to adapters would be interesting future work. In our revision, Appendix E.5 discusses this, and we compare our contrastive adapter approach to training an adapter with JTT more explicitly. We include results below for CLIP ResNet-50 embeddings, and find that contrastive adapters obtain higher worst-group (WG) acc. than JTT on 3 of 4 benchmarks (Table 17).
> >
> > |   Acc. (%)        |   Waterbirds WG   |     Waterbirds  Avg      |     CelebA WG     |       CelebA Avg      |  BREEDS Liv-17  WG |    BREEDS Liv-17  Avg        |   CIFAR-10.02 WG  |    CIFAR-10.02 Avg        |
> > |----------|:--------------:|:----------:|:--------------:|:----------:|:--------------:|:----------:|:--------------:|:----------:|
> > | JTT      |   69.2 ± 2.3   | 85.1 ± 0.4 |   86.3 ± 0.9   | 88.2 ± 0.4 |   61.4 ± 2.5   | 91.4 ± 0.3 | **63.5 ± 1.3** | 81.7 ± 0.2 |
> > | Ours     | **83.7 ± 0.7** | 89.4 ± 0.9 | **90.0 ± 0.4** | 90.7 ± 0.4 | **62.0 ± 1.6** | 90.9 ± 0.3 |   60.7 ± 1.7   | 80.9 ± 0.2 |
> >
> > Although not a perfect comparison, in Sec. 5.2 and Table 5 of our original submission we also compared to JTT-trained ResNet-50s (the standard JTT setup), and found our approach obtained comparable or better group-robustness while only training <1% of the parameters.
> >
> > **Concern with the “foundation model” term**
> > We 100% agree that these models are not “foundational”; our first contribution showing the poor zero-shot group-robustness of these models was very much motivated by similar sentiments to yours. We debated between “foundation model” and “large pretrained model”, but chose the former because of its coined connotation that these models can be unreliable foundations that need to be further developed or adapted for desired tasks [4], which we think agrees with your sentiment. We thought this aligned with our work as well, showing how these models can have poor group robustness, before proposing methods that build on these models to improve their robustness.
> >
> > However, we acknowledge the confusion with the term, its grammar, and usual definitions of “foundation”. We will gladly change the term in the final paper if you feel your concern is still not addressed.
> >
> > [1] Radford et al. Learning transferable visual models from natural language supervision. ICML 2021.
> > [2] Khosla et al. Supervised contrastive learning. NeurIPS 2020.
> > [3] Kirichenko et al. Last layer re-training is sufficient for robustness to spurious correlations. 2022.
> > [4] Bommasani et al. On the Opportunities and Risks of Foundation Models. 2022.

---

> > > ### Comment · Reviewer_2hnT · 2022-08-07
> > > **Reviewer's Response**
> > >
> > > I thank the authors for their efforts in addressing my questions. I am satisfied with the answers, and have gladly raised my score to 6. On the term of Foundation Model, I suggest the authors to change the term to avoid unnecessary controversy or diminishing the contribution of the manuscript, and then it will serve as a neutral scientific report on facts and facts only.

---

### Author Response · Authors · 2022-08-02
**General Response and Revision Updates**

We thank reviewers for their thoughtful reviews and insightful comments. Reviewers consistently appreciated the problem we presented, which was on the group robustness of large pretrained models. They noted its practical importance [2hnT, TAXS, 8jHt] and relevance to the hope of the community [2hnT].

Reviewers also appreciated our proposed method’s effectiveness as a solution, noting its improvements in robustness across models [2hnT], a large variety of scenarios [8jHt], and by a large margin over existing baselines [TAXS].

In our revision, we made several reviewer-suggested changes to better present and clarify the insights and details in the paper. These include:
1. Clarifying and extending existing ablations of our method’s components (App. E.2). These show that both the sampling procedure and contrastive objective of our method are integral for best group robustness for certain settings (Waterbirds, CelebA; where ERM adapters perform poorly), but that just the contrastive objective can also improve robustness in other settings (BREEDS Living-17, CIFAR-10.02; where ERM adapters perform well) (Table 14).


2. Comparing to and discussing additional baselines for efficiently improving group robustness (App. E.5). We adapt recent robustness methods (JTT, CNC, JM1) to train adapters in the pretrained model setting, and find contrastive adapters still obtain best worst-group accuracy on 3 of 4 benchmarks (Table 17).


3. Performing additional experiments to study how training data may impact our method’s relative performance (App. E.6). Our results suggest that datasets where ERM-trained adapters perform best are not necessarily more challenging for our method; rather, they have properties that make ERM training more competitive (e.g., small group shifts or balanced groups). Our method still significantly improves group robustness over zero-shot classification in these settings, and significantly improves robustness over ERM in harder settings with more significant distribution shifts between groups (Table 18) and more extreme group imbalances (Table 19).

Due to space constraints, most of these changes are detailed in the Appendix (now attached to the 9 page main paper / main submission file). We will include the key results in the main paper section of the final paper (following the author FAQs that an extra page is allowed).

We also clarify the scope and aims of our study, which (a) demonstrated poor group robustness in existing large pretrained models, (b) found that existing baselines (such as ERM-trained adapters) did not consistently improve group robustness, and (c) aimed to develop methods that could consistently & efficiently improve group robustness.

Please find our comments for individual reviewers in their respective threads below.

---

### Author Response · Authors · 2022-08-07
**Thanks for reviewer responses; 2nd revision updates**

Many thanks again to the reviewers and area chairs for presiding over our paper. We very much appreciate the helpful feedback, and are glad to hear that our responses have clarified and addressed the helpful questions of Reviewers 8jHt and TAXS. We are still hoping to hear from Reviewer 2hnT. We believe their reviews were particularly helpful for clarifying important baselines and paper details, and that we have addressed them in our initial response. We are happy to follow-up with any additional discussion.

Based on the helpful responses from Reviewers 8jHt and TAXS so far, we uploaded a new revision incorporating additional feedback. The changes include:
* Adding the ablation of our proposed method’s components to the main paper (now Section 5.2, previously Appendix E.2.2).
* Adding additional citations on out-of-distribution robustness, e.g., [31, 32, 62, 82]
* Adding a table of contents for navigating the appendix (page 17).

We thank the reviewers again for helping improve our paper. We apologize that this may break line references in our prior responses (changes from the original submission are still marked in blue).

Finally, as Reviewers 2hnT and TAXS brought up their preference for the term “large pretrained model” over “foundation model”, and 8jHt brought up modifying the paper title for clarity, we are working on making these changes for the final paper.

**References**
[31] Hendrycks & Dietterich. Benchmarking Neural Network Robustness to Common Corruptions and Perturbations. ICLR 2019.
[32] Hendrycks et al. Natural Adversarial Examples. CVPR 2021.
[62] Recht et al. Do ImageNet classifiers generalize to ImageNet? ICML 2019.
[82] Zhao et al. ROBIN: A Benchmark for Robustness to Individual Nuisances in Real-World Out-of-Distribution Shifts. ECCV 2022.

---

### Meta-Review · Area_Chair_et4r · 2022-08-26

**Recommendation:** Accept
**Confidence:** Certain

**Metareview:**

This paper received unanimous recommendations of acceptance from the reviewer. The authors did a good job addressing concerns from the reviewers, especially with the additional ablation studies to decouple the gains from other techniques such as SupCon. The AC agrees with the reviewers regarding the contribution of this paper and recommends acceptance.

**Award:**

No

---

### Decision · Program_Chairs · 2022-09-14

Accept